

# Ranaviruses and reptiles

Wytamma Wirth[1], Lin Schwarzkopf[2], Lee F. Skerratt[3] and Ellen Ariel[1]

[1] College of Public Health, Medical and Veterinary Sciences, James Cook University of North Queensland, Townsville, QLD, Australia
[2] College of Science and Engineering, James Cook University of North Queensland, Townsville, QLD, Australia
[3] Melbourne Veterinary School, Faculty of Veterinary and Agricultural Sciences, University of Melbourne, Melbourne, Australia

## ABSTRACT

Ranaviruses can infect many vertebrate classes including fish, amphibians and reptiles, but for the most part, research has been focused on non-reptilian hosts, amphibians in particular. More recently, reports of ranaviral infections of reptiles are increasing with over 12 families of reptiles currently susceptible to ranaviral infection. Reptiles are infected by ranaviruses that are genetically similar to, or the same as, the viruses that infect amphibians and fish; however, physiological and ecological differences result in differences in study designs. Although ranaviral disease in reptiles is often influenced by host species, viral strain and environmental differences, general trends in pathogenesis are emerging. More experimental studies using a variety of reptile species, life stages and routes of transmission are required to unravel the complexity of wild ranavirus transmission. Further, our understanding of the reptilian immune response to ranaviral infection is still lacking, although the considerable amount of work conducted in amphibians will serve as a useful guide for future studies in reptiles.

## INTRODUCTION

Ranaviruses (family *Iridoviridae*) are emerging lethal pathogens of ectothermic vertebrates. First discovered in 1965 (*Granoff, Came & Rafferty, 1965*), ranaviruses were initially studied for their interesting molecular biology but rose to reportable pathogen status as more epizootics were discovered (*Schloegel et al., 2010*; *Gray & Chinchar, 2015*). The vast majority of research on the genus *Ranavirus* has been conducted in amphibians (*Rana* is Latin for frog), but despite their name, ranaviruses do not occur only in amphibians (*Chinchar & Waltzek, 2014*). This group of viruses infects over 175 species of ectothermic vertebrates; including reptile species from at least 12 different families (*Duffus et al., 2015*). Temperature appears to be the major factor preventing ranaviral infection outside of ectothermic vertebrates; these viruses can replicate in mammalian cell lines, but only when the temperature is below 32 °C (*Gray & Chinchar, 2015*).

Many advances in the field of ranavirology have been made since the discovery of ranaviruses; however, for the most part, this research is specific to amphibians. Reptiles and amphibians are very different physiologically and although they sometimes share habitats, their ecology is different. Some results from one host group can translate to the

Corresponding author
Wytamma Wirth,
wytamma.wirth@my.jcu.edu.au

other; however, there is no substitute for host-specific research. As ranavirus research continues, it is important to focus efforts on all Classes of hosts, including reptiles.

Since the initial report of ranaviruses in reptiles in the early 1980s (*Heldstab & Bestetti, 1982*), infections have been reported in wild and captive reptiles, and the number of reports continues to grow, representing an increasing problem for reptiles (*Marschang, Stöhr & Allender, 2016*). In this review, we summarise findings in all areas of reptilian ranavirus research. We identify major gaps in this field of knowledge and include recommendations for future research directions.

## SURVEY METHODOLOGY

To ensure this review included as many publications focusing on ranaviruses and reptiles as possible, an extensive search of multiple databases using broad search queries was conducted. Databases used in the search strategy included: Web of Science, PubMed, and Google Scholar. The search strategy included keywords such as 'ranavirus' and 'reptiles' and their conjugations as well as more specific terms such as 'turtle', 'lizard', and 'snake'. To broaden the search further, references for articles found in the initial database search were then assessed for content relating to ranaviruses and reptiles. As a baseline for general ranavirus literature, relevant references were extracted from the 2015 *Ranavirus* book (*Gray & Chinchar, 2015*).

## TAXONOMY

Ranaviruses are large (~150 nm), nucleocytoplasmic viruses with icosahedral virions and double-stranded DNA genomes that contain approximately 100 genes (*Jancovich et al., 2015*). *Ranavirus* is a genus in the family *Iridoviridae*: a group of five related viral genera. Of the five *Iridoviridae* genera, only ranaviruses cause significant disease in wild reptiles.

The taxonomy of the genus *Ranavirus* is changing; as more viruses are isolated and sequenced a clearer picture of the phylogenetic distribution of this group is developing. The international committee on taxonomy of viruses (ICTV) currently recognises eight species in the genus *Ranavirus* (*Lefkowitz et al., 2018*), none of which were originally isolated from reptiles. The official ICTV process of species recognition takes time and coordination within the scientific community. Many isolates, including isolates from reptiles, remain unclassified (*Chinchar et al., 2017*).

The current phylogeny of the genus *Ranavirus* can be subdivided into five major lineages based on comparison of conserved genes (*Claytor et al., 2017*; *Jancovich et al., 2015*; *Stöhr et al., 2015*; *Price et al., 2017*). No *Ranavirus* lineage exclusively infects reptiles, and the majority of reptile infections appear to originate from putative amphibian specialist viruses (*Price et al., 2017*). The factors that control the host specificity of these viruses remain unknown. Phylogenetic analyses of sequences from different reptilian and amphibian viruses have revealed that viruses found in reptiles are often more closely related to amphibian ranaviruses from the same geographical region than to each other (*Stöhr et al., 2015*). This provides support for the hypothesis that the jump into reptile hosts is relatively recent and has occurred multiple times (*Jancovich et al., 2010*, *Stöhr et al., 2015*).
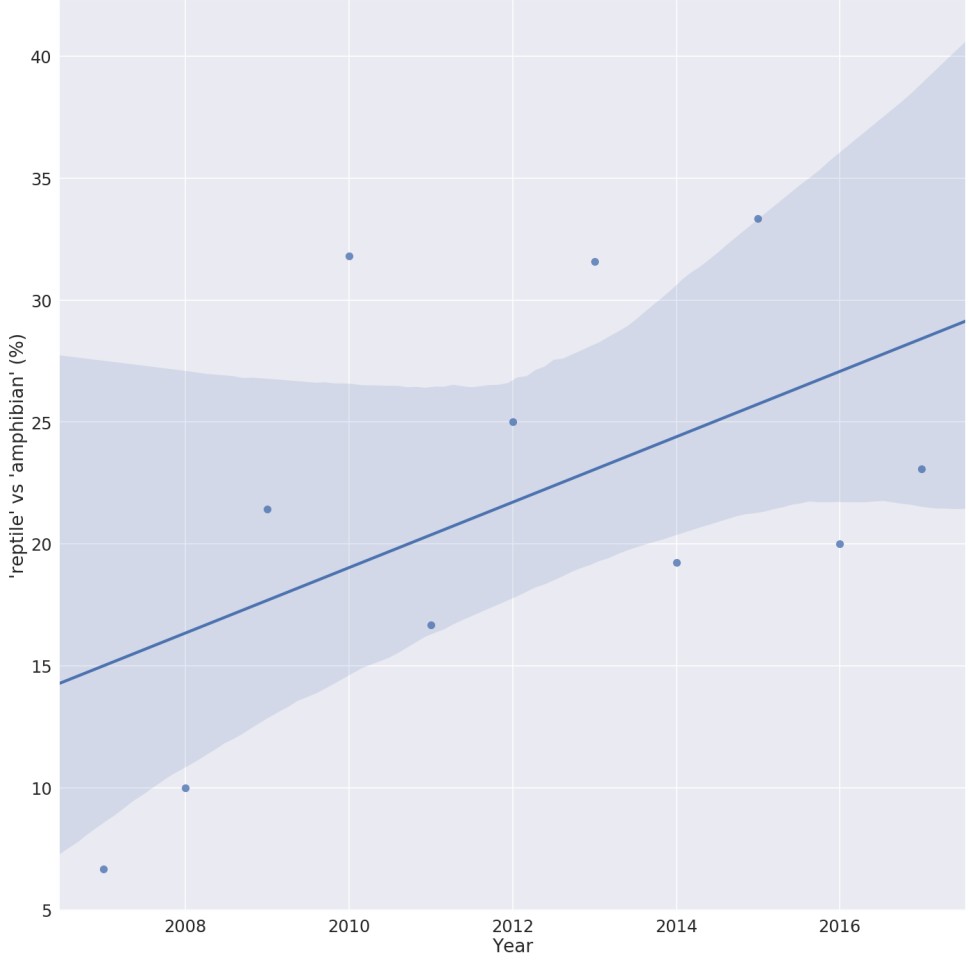

**Figure 1** **Trend in the number of ranavirus papers referring to reptiles.** Ratio of ranavirus papers ($n =$ 449) using the terms 'reptile' vs 'amphibian' in their title or abstract, showing the increase in the relative percent of publications referring to reptiles. A value of 100% would indicate the same number of Ranavirus papers use the term 'reptile' as 'amphibian'. Solid line is the linear trend line fitted with 95% confidence interval (shaded area).

## BIBLIOMETRICS

Despite a lack of host specificity, the vast majority of ranaviral literature is on amphibians. As of February 2018, 449 references were returned when the Web of Science ™ database was queried for the topic 'ranavirus'. Of these, over 200 used the term 'amphibian' in their title or abstract while fewer than 60 used the term 'reptile'. However, plotting the usage of these terms over the last 10 years shows a steady increase in the ratio of 'reptile' to 'amphibian' terms, possibly reflecting an increase in reptilian ranavirus research or an increased awareness of the role of reptiles in this disease (Fig. 1).

## DIAGNOSTICS AND SURVEILLANCE

The World Organization for Animal Health (OIE) provides guidelines for diagnostic methods in their Diagnostic Manual for Aquatic Animal Health (*OIE, 2012*), and *Miller et al. (2015)* summarised the diagnostic techniques used in ranaviral research. The most commonly used methods to confirm the presence of a ranavirus in host samples have included electron microscopy, enzyme-linked immunosorbent assays (ELISAs), viral isolation, immunohistochemistry (IHC), DNA amplification using polymerase chain reaction (PCR), and more recently, next generation sequencing. All of these techniques have been used at some stage in the study of ranaviruses from reptiles; however, the selection of diagnostic technique is highly dependent on the resident expertise in the laboratory, the data required, and the type of study (*Miller et al., 2015*). Before their application in new hosts or against new pathogens, all diagnostic techniques should be thoroughly tested and optimised with appropriate controls (e.g., different species of reptiles or ranaviruses, or both) (*Wobeser, 2007*).

Most ranaviruses can be grown using commercially available fish cell lines (*Miller et al., 2015*). Reptile cell lines such as Russell's viper heart cells, gecko lung cells, turtle heart, and iguana heart cells have also been used successfully to isolate ranaviruses from reptiles (*Hyatt et al., 2002*; *Johnson et al., 2008*; *Alves de Matos et al., 2011*).

### Serology

Serological surveys, employing various ELISAs, have been used to assay reptiles for anti-ranaviral antibodies (*Johnson et al., 2010*; *Ariel et al., 2017*). Although these ELISA-based surveys have successfully detected anti-ranaviral antibodies in wild and captive chelonians and experimentally infected reptiles, the utility of such surveys is not clear due to the inconsistency of sero-conversion after ranaviral infection in reptiles. A captive group of chelonians, with a history of iridovirus outbreak, had a low proportion of seropositive individuals, and wild populations of North American chelonians were shown to have low sero-prevalence (*Johnson et al., 2010*). Experimentally-infected Australian reptiles do not show consistent patterns of sero-conversion, although wild populations can have high levels of antibodies (*Ariel et al., 2017*). Sero-surveys of large aquatic reptiles, such as freshwater turtles, could be useful as an indication of ranavirus occurring in freshwater environments but they would likely underestimate the true prevalence of exposure unless the accuracy of the test is determined (*Ariel et al., 2017*; *Johnson et al., 2010*). Exposed animals may fail to sero-convert or die before they are surveyed. Reptile antibody titres vary seasonally (more antibodies are produced in the warmer months), which must be taken into account when determining sensitivity and specificity cut-off values for diagnostic tests (*Wobeser, 2007*; *Zimmerman, Vogel & Bowden, 2010*; *Meddings, 2011*). Seasonal variation of anti-ranaviral antibodies has not been assessed. Using total IgY levels as an internal control may minimise diagnostic errors resulting from seasonal variations in antibody levels.

### Polymerase chain reaction

PCR-based assays have been used conventionally and in quantitative real-time assays to detect reptilian ranaviral DNA in a number of sample types including blood, oral and cloacal

swabs, and fresh and fixed tissues (*Pallister Gould et al., 2007*; *Allender, Bunick & Mitchell, 2013*; *Allender et al., 2013a*; *Goodman, Miller & Ararso, 2013*; *Butkus et al., 2017*; *Leung et al., 2017*; *Maclaine et al., 2018*). Molecular surveys of turtle populations for ranavirus have revealed that swabs and blood samples are not equally valid targets for ranavirus detection (*Allender et al., 2013a*). *Goodman, Miller & Ararso (2013)* also found that oral cloacal swabs were not as effective for ranavirus detection when compared with tail clip tissue samples. Given possible differences in sample type sensitivity, it would seem advisable to collect multiple samples (e.g., both blood and swabs) when conducting a molecular survey for reptilian ranaviruses (a method employed in many studies). It is also possible to use bone marrow as a source of DNA for ranavirus detection from reptile carcases in which other viable tissue samples may have decayed (*Butkus et al., 2017*).

The preferred target of ranaviral PCR assays is the major capsid protein (MCP) gene as it is highly conserved throughout the ranaviral lineage (*Miller et al., 2015*). Because of the multispecies host range of most reptile ranaviruses, reptile-specific PCR assays are not required. Many different PCR assays have been used in reptile studies; generally the assay of choice depends on the question asked. PCR assays with broad targets such as those from *Mao, Hedrick & Chinchar (1997)* have been used in surveys (*Goodman, Miller & Ararso, 2013*) and more specific, probe-based PCR assays have also been used in surveys of reptiles (*Allender et al., 2013a*). Probe-based assays generally sacrifice broad detection range for increased sensitivity (*Allender, Bunick & Mitchell, 2013*). Sensitive probe-based assays have also been used in experimental infections with a known viral target to determine levels of infection (*Allender et al., 2013b*). A new qPCR assay developed by *Leung et al. (2017)* should provide more accurate viral load determination by using an internal host control DNA target conserved in reptiles. Large product PCR assays have been used in phylogenetic studies of reptilian ranaviruses (*Stöhr et al., 2015*). As the cost of sequencing decreases, it is becoming increasingly popular to use high throughput sequencing methods to more accurately identify and characterise viral isolates (*Hick et al., 2016*; *Subramaniam et al., 2016*).

Environmental DNA (eDNA) -based detection may be an effective method for assessing the presence of ranaviral DNA in populations of aquatic reptiles. Ranavirus outbreaks in aquatic frog populations have been detected using eDNA PCR methods (*Hall et al., 2016*). Aquatic reptiles (Testudines) with ranaviral infections can shed virions into their surroundings, indicating that eDNA detection may be possible, although no publication has yet confirmed this *in situ* (*Brenes et al., 2014a*). Mosquitoes may be useful targets for detecting ranaviruses in reptile populations. *Kimble et al. (2014)* found ranaviral DNA in mosquitoes associated with a box turtle epizootic. Ranaviral PCR testing of mosquitoes (xenosurveillance) could be combined with DNA barcoding to determine the origin of the mosquito blood meal as well as the presence of ranavirus (*Bitome-Essono et al., 2017*).

## Antigen assays

Immunohistochemistry (IHC) has been successfully used in reptilian ranaviral studies to visualise the location of the viral protein in tissue samples (*Hyatt et al., 1991*; *Ariel et al., 2015*). Unlike PCR, IHC targets proteins in histological sections of tissues. Ranaviral IHC

assay results combined with histopathology can be used to correlate pathogenesis with the location of viral antigens (*Becker et al., 2013*; *Ariel et al., 2015*; *De Jesús Forzán et al., 2015*). *Ariel et al. (2015)* used cross-reactive polyclonal anti-EHNV antibodies to detect BIV antigens in infected turtle tissues. They found IHC staining associated with vascular endothelial cells, possibly indicating that viraemia preceded the systemic infection observed in these animals.

Another antigen assay has used anti-ranaviral monoclonal antibodies in a double antibody sandwich ELISA to detect viral particles in soft-shelled turtles (*Zhang et al., 2010*). In this case, the virion was detected with 98% specificity when compared with conventional PCR as the gold standard.

Because of the variability in ranaviral disease signs and severity within and among reptile species (see pathology section), suspected cases of ranaviral disease must be confirmed with laboratory diagnostic techniques. Epidemiological surveys must be adequately designed and powered to ensure ranavirus prevalence is accurately reported (*Gray & Chinchar, 2015*). There have been several studies reporting the negative results of epidemiological surveys (*Hanlon et al., 2016*; *Kolesnik, Obiegala & Marschang, 2017*; *Winzeler et al., 2018*). These results are extremely valuable as they also help describe the distribution and emergence patterns of reptilian ranaviruses; however, it is important to consider the sampling protocols and diagnostic choice when evaluating and comparing epidemiological studies (*Gray & Chinchar, 2015*).

## DISTRIBUTION, HOST RANGE AND IMPACT

Ranaviruses capable of infecting reptiles have been found on all continents, except Antarctica (*Duffus et al., 2015*). Ranaviruses have been detected in over 12 families of the orders Testudines (turtles, tortoises and terrapins) and Squamata (lizards and snakes). It is important to note that given reptile populations often share habitat with susceptible fish and amphibian species, it may be possible to infer reptile ranavirus distributions based on amphibian ranaviral prevalence patterns and *vice versa* (*Duffus et al., 2015*). It is not clear if reptilian ecology influences patterns of ranavirus host range or susceptibility. Aquatic reptiles may be more likely to be exposed to ranaviruses; however, ranaviruses are still detected in terrestrial reptiles (*Duffus et al., 2015*). There is some evidence aquatic turtles are less susceptible to ranaviral disease; however, this is far from settled and should be investigated further (*Brunner et al., 2015*). Recently, *Adamovicz et al. (2018)* reported that the use of moist microhabitats is correlated with ranavirus detection in eastern box turtles.

### Testudines (turtles, tortoises and terrapins)

Koch's postulates have been confirmed in Testudines with infection and disease demonstrated in red-eared sliders (*Trachemys scripta elegans*) and box turtles (*Terrapene ornata ornata*) infected with a Burmese star tortoise *Ranavirus* isolate (*Johnson, Pessier & Jacobson, 2007*). The first reported cases of ranaviral infections in Testudines were identified microscopically during the 1980s in Hermann's tortoises (*Testudo hermanni*) in Switzerland (*Heldstab & Bestetti, 1982*). Following this, ranaviruses were predominantly

isolated from box turtles (*Terrapene carolina*) and were identified as the aetiological agent of 'red neck disease' in the soft-shelled turtle (*Pelodiscus sinensis*) (*Chen, Zheng & Jiang, 1999*). In the last decade, several new reports of ranaviral infections in Testudines have been published (*Johnson et al., 2008*; *Johnson et al., 2010*; *Belzer & Seibert, 2011*; *Allender, 2012*; *Stöhr et al., 2015*; *Perpiñán et al., 2016*; *Butkus et al., 2017*; *Agha et al., 2017*; *Archer et al., 2017*; *Adamovicz et al., 2018*). Despite the increasing number of reports of infections in the Testudines, ranaviral disease in these reptiles is still likely to be underreported due to a lack of awareness, an incomplete understanding of the pathology caused by the disease, few long-term studies, and minimal population monitoring (*Duffus et al., 2015*). Sea turtles are a group of reptiles that have received little attention from ranavirus researchers, despite the existence of ranavirus infections in marine fish (*Whittington, Becker & Dennis, 2010*).

## Squamata (lizards and snakes)

The first reports of ranaviral infection in squamates came after several green tree pythons were seized during an attempt to illegally import them into Australia from Indonesia. *Hyatt et al. (2002)* reported that these snakes were infected with an FV3-like ranavirus isolate. In 2005, *Marschang, Braun & Becher (2005)* reported the first ranaviral infection in lizards. The reports of ranavirus infections in squamates have been, for the most part, restricted to groups of captive lizards, providing little evidence of the role of ranavirus infection in wild squamate populations (*Stohr et al., 2013*; *Behncke et al., 2013*; *Stöhr et al., 2015*; *Tamukai et al., 2016*). Although no epizootics have been reported, ranaviral DNA and seropositive animals have been detected in wild squamate populations (*Alves de Matos et al., 2011*; *Ariel et al., 2017*; *Goodman, Hargadon & Carter, 2018*).

## Rhynchocephalia (tuatara), Archosaurs (crocodiles, birds)

There have been no documented cases of ranavirus infections in animals from the other groups of the class Reptilia, namely the Rhynchocephalia (tuatara) and the archosaurs (crocodilians and birds). The tuatara only inhabit parts of New Zealand, and although ranaviruses are believed to be present (i.e., short-finned eel ranavirus, *Bovo et al., 1993*), no studies have been published on the presence of ranavirus in tuatara. While yearling Australian freshwater crocodiles (*Crocodylus johnstoni*) were exposed to ranavirus (BIV) under laboratory conditions, this challenge did not cause any adverse effects in the yearlings and the virus could not be re-isolated (*Ariel et al., 2015*). A serosurvey of wild freshwater crocodiles did show evidence of anti-ranaviral antibodies, indicating that wild populations are likely exposed (*Ariel et al., 2017*). It is important to continue to study apparently resistant species, like crocodiles, as they may give insights into the determinants of immunity. Birds and reptiles are closely related; crocodiles are genetically more closely related to birds than they are to lizards. There are no reports of birds infected by ranaviruses (this is probably related to endothermy); despite this, birds may still play a role in ranaviral transmission. It has been hypothesised that migratory birds, acting as mechanical vectors, are responsible for some of the geographic transmission of ranaviruses (*Whittington et al., 1996*).

## PATHOLOGY

The clinical signs and pathogenesis of natural ranaviral infection in reptiles can be extremely variable. Mortality during an epizootic can range from 0–100% and the effect on a host can vary from quite mild to extremely severe, requiring immediate veterinary attention or euthanasia (*Miller et al., 2015*). There is evidence that reptiles can also be asymptomatic carriers of ranaviruses (*Stohr et al., 2013*; *Goodman, Hargadon & Carter, 2018*). Quiescent viral reactivation in amphibians that have recovered from infection is possible; however, the same is not known for reptiles (*Robert et al., 2014*). The complex presentation and inconsistency in the pathogenesis of ranaviral infection in reptiles may occur because of the influence of host physiology and life history, and varying degrees of viral virulence, stressors, and temperatures acting on the course and outcome of infection (see Susceptibility section).

Descriptions of pathogenesis in reptiles infected with a variety of ranaviral strains in several host species under experimental, wild, and captive conditions are presented in Table 1. Despite differences in descriptions of pathogenesis and the fact the reports are often confounded with co-infections (*Sim et al., 2016*; *Adamovicz et al., 2018*), some common patterns of ranaviral pathogenesis have emerged.

General lethargy and inappetence are associated with many cases of ranaviral infection in reptiles; however, such clinical signs are common to many diseases and are not pathognomonic for ranaviral infection. Turtles often present with respiratory signs, including nasal and oral discharge (*Johnson, Pessier & Jacobson, 2007*; *Johnson et al., 2008*; *Allender et al., 2013b*; *Kimble et al., 2017*). Oedema, especially of the eyes or neck, is also commonly associated with this infection in the order Testudines (*Chen, Zheng & Jiang, 1999*; *Johnson, Pessier & Jacobson, 2007*; *Johnson et al., 2008*; *Allender et al., 2013b*). The clinical signs of ranaviral infection in Squamates are scarcely described. This is partially due to the lack of experimental infection trials in this group, which would help describe pathogenesis markers. *Maclaine et al. (2018)* recently demonstrated the susceptibility of an Australian lizard species (*Intellagama lesueurii lesueurii*) to ranaviral infection, documenting that clinical signs and histopathological changes varied with inoculation route. With increasing descriptions of ranaviral infected lizards over the last decade, an emerging trend suggests that skin lesions may be a common occurrence (*Behncke et al., 2013*; *Stohr et al., 2013*; *Tamukai et al., 2016*).

Ranaviral infections are systemic, and there is often extensive damage to multiple organs during infection, especially the liver and spleen in reptiles. Liver lesions are also very common in the pathogenesis of ranaviruses in amphibian and fish species (*Miller et al., 2015*). Histopathology is frequently characterised by inflammation and multifocal necrosis in multiple organs, and is often associated with hematopoietic tissue (*Ariel et al., 2015*). Reptilian hosts of ranaviruses experience a range of histological changes including necrosis and inflammation of the respiratory tract, pneumonia, conjunctivitis, stomatitis, esophagitis, tracheitis, necrosis of endothelial cells and the submucosa of the gastrointestinal tract, glomerulonephritis, multifocal hepatic necrosis, splenitis, intracytoplasmic inclusion bodies in many tissues, and necrotizing myositis (see Table 1). Evidence from epizootics in reptiles indicates that ranaviral infection can be accompanied by secondary pathogens

*Peer*J

**Table 1  Representative reptilian ranaviral pathogenesis.** This table includes only cases of moribund reptiles where sufficient clinical description was given.

| Reference | Order | Family | Genus | Species | Population | Behaviour | Oral | Nasal | Ocular | Skin | Other signs | Skin | Oral cavity | Gastrointestinal tract | Upper respiratory tract | Lower respiratory tract | Liver | Spleen | Kidney | Pancreas | Muscle |
|---|---|---|---|---|---|---|---|---|---|---|---|---|---|---|---|---|---|---|---|---|---|
| | | | | | | Clinical signs | | | | | | Pathogenesis | | | | | | | | | |
| *Behncke et al. (2013)* | Squamata | Agamidae | *Japalura* | *splendida* | Captive/Wild | CN, DA | | | | L | | | | | | | H, N | | N | | |
| *Stöhr et al. (2013)* | Squamata | Agamidae | *Pogona* | *vitticeps* | Captive | | | | | L | | I | | | | | | | | | I |
| *Tamukai et al. (2016)* | Squamata | Agamidae | *Pogona* | *vitticeps* | Captive | DA | | | | L, U | | I, N, U | | | | | | | | | |
| *Maclaine et al. (2018)* | Squamata | Agamidae | *Intellagama* | *lesueurii lesueurii* | Experimental | An, CN, DA | | | | L, U | Abdomen swelling | N | | | H, I | | N | H, I, N | N | | |
| *Stöhr et al. (2013)* | Squamata | Anguidae | *Ophiosaurus* | *gracilis* | Captive | | | | | L | | I, U | | | | | | | | | |
| *Stöhr et al. (2013)* | Squamata | Dactyloidae | *Anolis* | *sagrei* | Captive | DA | | | | L | | I, N | | | | | | | | | |
| *Stöhr et al. (2013)* | Squamata | Dactyloidae | *Anolis* | *carolinensis* | Captive | | | | | I, U | | I, U | | | | | | | | | |
| *Marschang, Braun & Becher (2005)* | Squamata | Gekkonidae | *Uroplatus* | *fimbriatus* | Captive | An | | | | | | | I, N, U | | | | | N | I | | |
| *Stöhr et al. (2013)* | Squamata | Iguanidae | *Iguana* | *iguana* | Captive | | | | | L | | | | | | | | | | | |
| *Hyatt et al. (2002)* | Squamata | Pythonidae | *Morelia* | *viridis* | Captive/Wild | An, DA | U | | | | | N | I | I, N | I | | N | N | N | | N |
| *Duffus et al. (2015)* | Squamata | Pythonidae | *Python* | *brongersmai* | Captive | | | | | | | | I | | | | I | | | | |
| *Ariel et al. (2015)* | Testudines | Chelidae | *Emydura* | *macquarii krefftii* | Experimental | An, DA | | | | | | | | I, N | | | I, N | I, N | I, N | I, N | I, N |
| *Johnson et al. (2008)* | Testudines | Emydidae | *Terrapene* | *carolina bauri* | Wild | | | | D | D, E | | | I | | | | | I | | | |
| *Johnson et al. (2008)* | Testudines | Emydidae | *Terrapene* | *carolina carolina* | Captive/Wild | | | | D | D, E | | | I | I | I | | | I | | | |
| *Johnson et al. (2008)* | Testudines | Emydidae | *Terrapene* | *carolina carolina* | Wild | | | | | D, E | Aural abscesses | | I | | I | | | I | I | | |
| *Johnson et al. (2008)* | Testudines | Emydidae | *Terrapene* | *carolina carolina* | Wild | | | | | D | | N | | | I | | | I | | | |
| *Johnson et al. (2008)* | Testudines | Emydidae | *Terrapene* | *carolina carolina* | Captive | | | | | | | | | | | | | I | | | |

**Table 1** (*continued*)

| Reference | Order | Family | Genus | Species | Population | Behaviour | Oral | Nasal | Ocular | Skin | Other signs | Skin | Oral cavity | Gastrointestinal tract | Upper respiratory tract | Lower respiratory tract | Liver | Spleen | Kidney | Pancreas | Muscle |
|---|---|---|---|---|---|---|---|---|---|---|---|---|---|---|---|---|---|---|---|---|---|
| | | | | | | Clinical signs | | | | | | Pathogenesis | | | | | | | | | |
| *DeVoe et al. (2004)* | Testudines | Emydidae | *Terrapene* | *carolina carolina* | Captive/Wild | An, DA | | | C | A, U | Respiratory distress | I | N | I | | | I | I | I | I | I |
| *Allender et al. (2006)* | Testudines | Emydidae | *Terrapene* | *carolina carolina* | Wild | An, DA | L | D | C, D | | Weight loss | I, N | I, N | I,N | N | | | I | I, N | | |
| *Johnson, Pessier & Jacobson (2007)* | Testudines | Emydidae | *Terrapene* | *ornata ornata* | Experimental | An, DA | | | D | | | | | | | | | L | | | |
| *Johnson, Pessier & Jacobson (2007)* | Testudines | Emydidae | *Trachemys* | *scripta elegans* | Experimental | An, DA | | D | C, D | | Increased basking, Exophthalmus, hyphema | I | H, I | | | T | N, T | L | T | | |
| *Allender et al. (2013b)* | Testudines | Emydidae | *Trachemys* | *scripta elegans* | Experimental | DA | L | D | D | A | Leg swelling | | N, U | I, T | | | L, T | I, N | | | I, N |
| *Benetka et al. (2007)* | Testudines | Testudinidae | *Stigmochelys (Geochelone)* | *pardalis* | Captive | An, DA | D | I | | | | | I, N | | | I | | | | | |
| *Marschang et al. (1999)* | Testudines | Testudinidae | *Testudo* | *hermanni* | Captive | | | | | | | | I, N | | | I, N | | | | | |
| *Blahak & Uhlenbrok (2010)* | Testudines | Testudinidae | *Testudo* | *hermanni* | Captive | | | | | | Emaciation | | I | | | | | | | | |
| *Blahak & Uhlenbrok (2010)* | Testudines | Testudinidae | *Testudo* | *kleinmanni* | Captive | | | | | | | | I | | I | I | | | | | |
| *Blahak & Uhlenbrok (2010)* | Testudines | Testudinidae | *Testudo* | *marginata* | Captive | | | | | | | | I | | | N | | | | | |
| *Heldstab & Bestetti (1982)* | Testudines | Testudinidae | *Testudo* | *hermanni* | Captive | | | | | | | | I, N | | | | I, N | I, N | | | |
| *Johnson et al. (2008)* | Testudines | Testudinidae | *Geochelone* | *platynota* | Captive | | | D | C | | Neck swelling | I | I, N | | | | | | I | I | |
| *Johnson et al. (2008)* | Testudines | Testudinidae | *Gopherus* | *polyphemus* | Wild | | | D | C, D, E | | | | I | | | | | | I | I | |
| *Westhouse et al. (1996)* | Testudines | Testudinidae | *Gopherus* | *polyphemus* | Wild | DA | | D | D | | Respiratory disease | | I | | | I, N, U | | | | | |
| *Chen, Zheng & Jiang (1999)* | Testudines | Trionychidae | *Pelodiscus* | *sinensis* | Captive/ Experimental | | | | | | Red neck, neck swelling | H | | | | | H | | | | |

**Notes.**

A, abscess; An, anorexia; C, conjunctivitis; CN, central nervous disorders; DA, decreased-activity/depression/lethargy; D, discharge; E, oedema; H, haemorrhage; I, inflammation; L, lesion; N, necrosis; T, thrombi; U, ulceration.

that may exacerbate the disease and mask clinical signs of ranaviral infection (*Stohr et al., 2013*; *Sim et al., 2016*; *Archer et al., 2017*).

## TRANSMISSION

The natural route of transmission of ranaviruses in wild populations of reptiles is still debated, although experimental data suggest multiple transmission routes are possible (*Brunner et al., 2015*). During an experimental challenge of adult red-eared sliders (*Trachemys scripta elegans*), *Johnson, Pessier & Jacobson (2007)* found that the orally exposed animals were refractory to infection while animals challenged with the same dose via intramuscular injection developed severe disease. In another study, exposure to ranavirus in water *via* cohabitation resulted in subclinical infection in some red-eared slider hatchlings (*T. scripta elegans*), although the route of infection was not determined, and the concentration of virus in the water was not quantified (*Brenes et al., 2014a*). *Ariel et al. (2015)* found that adult freshwater turtles (*Emydura krefftii* and *Elseya latisternum*), freshwater crocodiles (*C. johnstoni*), and several species of snakes were refractory to infection irrespective of the route of exposure. The hatchlings of both species of freshwater turtles were susceptible to infection *via* intra-coelomic exposure although oral inculcation was not attempted. Juvenile Australian eastern water dragons (*Intellagama lesueurii lesueurii*) developed ranaviral disease from all exposure routes tested (oral, intramuscular, and cohabitation) (*Maclaine et al., 2018*).

Differences in susceptibility *via* different routes of exposure may reflect real differences in natural transmission routes between reptiles and other Classes. More experimental studies using a variety of species, life stages, and routes of transmission are needed to resolve this.

Amphibians are highly susceptible to ranaviral infection *via* all tested forms of inoculation (water bath, skin contact, oral inoculation or injection) (*Miller et al., 2015*). Fish are also susceptible *via* multiple inoculation routes, although it appears to be species-dependent (*Bang Jensen, Ersbøll & Ariel, 2009*; *Gobbo et al., 2010*; *Jensen et al., 2011*). Differences in viable transmission routes result in different epidemiologies, and thus research from other host classes with different viable transmission routes may not accurately reflect risks and susceptibility of reptilian populations. It is therefore important to account for variation in transmission routes among reptile species when developing statistical models for reptilian disease.

## VECTORS

Humans are contributing to the global spread of ranaviruses, primarily though global animal trade (*Kolby et al., 2014*; *Duffus et al., 2015*; *Stöhr et al., 2015*). Although there are reports of ranaviral infection in traded reptiles (*Hyatt et al., 2002*; *Stohr et al., 2013*), no systematic survey of ranaviral infection in traded reptiles has been conducted. There have been some ranaviral disease outbreaks in private reptile collections and zoos (*Marschang, Braun & Becher, 2005*; *Sim et al., 2016*), but the full extent of disease prevalence is hard

to assess, both because of inapparent infections, and lack of reporting of dead animals amongst reptile breeders and collectors.

Ranaviral DNA sequences have been identified in mosquitoes associated with a ranavirus outbreak in box turtles, providing evidence for vector transmission (*Kimble et al., 2014*). Ranaviral DNA and antigens have been detected in blood and blood-associated tissues of reptiles (*Allender et al., 2013a*; *Ariel et al., 2015*; *Miller et al., 2015*). Leeches are common ectoparasites of aquatic reptile species and can act as vectors for blood-borne diseases (*Siddall & Desser, 1992*; *Watermolen, 1996*; *Readel, Phillips & Wetzel, 2008*). There has been at least one report of a ranavirus-positive leech (PCR for MCP) associated with an infected amphibian host, although there are no reports for leeches of reptiles (*Hardman et al., 2013*). Some low density reptile populations that experience ranaviral epizootics do not appear to be capable of propagating ranaviral disease through physical contact alone (*Brunner et al., 2015*). Despite these indicators of the possible involvement of vectors in ranavirus transmission, no experimental studies have been published that support or refute this hypothesis in reptiles.

## RESERVOIRS

Ranaviral virions are extremely stable in controlled settings, they are capable of withstanding high and low pH and temperatures and are resistant to desiccation, remaining viable for days to years (*Granoff, Came & Rafferty, 1965*; *Langdon et al., 1986*; *Langdon, 1989*; *Munro et al., 2016*; *Nazir, Spengler & Marschang, 2012*). These qualities of stability may not to carry to ecological settings as interactions with the aquatic biotic communities can reduce the longevity of infectious ranaviral particles (*Brunner et al., 2015*). *Reinauer, Bohm & Marschang (2005)*, found that tortoise ranaviruses remain infectious in lake water samples and in soil for many days; however, biotic communities were not quantified. It also appears that moisture is important for persistence in soil environments (*Brunner et al., 2015*; *Nazir, Spengler & Marschang, 2012*). Animals, both live and dead, are also probably reservoirs for reptilian ranavirus infections (*Gray, Miller & Hoverman, 2009*). Reptiles are known to consume frogs, fish, and even other reptiles as a part of their natural diet (*Kischinovsky, Raftery & Sawmy, 2017*). Dead and decaying animals continue to release virions and might be consumed by susceptible reptiles (*Brunner et al., 2015*; *Gray & Chinchar, 2015*). Asymptomatic amphibians are sometimes reservoirs; they can spread virus to other susceptible species, and possibly reptiles, in multispecies ranavirus epizootics (*Brenes et al., 2014a*; *Brenes et al., 2014b*; *Brunner et al., 2015*).

## CORRELATES OF SUSCEPTIBILITY

Reptiles are ectotherms and so their physiology is strongly influenced by the temperature of their surrounding environment. By extension, the innate and adaptive immune response of reptiles is also linked to available environmental temperatures (*Zimmerman, Vogel & Bowden, 2010*). Ranavirus-infected reptiles, such as turtles, exhibit temperature-dependent pathogenesis (*Allender et al., 2013b*; *Allender et al., 2018*) similar to that observed in fish and amphibians (*Brunner et al., 2015*; *Brand et al., 2016*); however, the replication efficacy of

the virus is also linked to temperature (*Ariel et al., 2009*). Thus, it is difficult to determine the degree to which temperature-dependent pathogenesis is a result of the effect of temperature on the replication of the virus or on the immune system of the turtles. Several studies have quantified the temperature-dependent activity of the innate immune system of reptiles (*Merchant et al., 2006*; *Ferronato et al., 2009*; *Merchant et al., 2012*). In experimental infections of ranaviruses, temperature is often uncontrolled (reported as 'room temperature').

*Allender et al. (2013b)* suggested that an environmental temperature increase of 6 °C is enough to significantly reduce ranavirus loads and halve morbidity in infected adult turtles. However, in a follow up study, in juvenile turtles, it was found that increased temperature reduced median survival time of all four species tested (*Allender et al., 2018*). Similar patterns of reduced time until death but lower mortality rates with increasing temperature have been seen with other environmental temperature-dependent host-pathogen systems such as amphibians with chytridiomycosis (*Berger et al., 2004*). This pattern of temperature-related susceptibility in reptiles is important for future studies to quantify.

The effects of stressors on reptilian ranaviral disease are poorly understood (*Polakiewicz & Goodman, 2013*). Several studies in amphibians have examined the effects of stressors on disease in experimental infections (*Echaubard et al., 2010*; *Forson & Storfer, 2006*; *Haislip et al., 2012*; *Kerby, Hart & Storfer, 2011*; *Reeve et al., 2013*), and epidemiological studies have looked for correlations between environmental stressors and ranaviral prevalence (*St-Amour et al., 2008*; *Brunner et al., 2015*). The immunosuppressive effects of some anthropogenic stressors (e.g., pesticides, herbicides, and heavy metals) on the reptile immune system suggest a possible mechanism of environmental influence on susceptibility. Future epidemiological studies should consider these factors (*Keller et al., 2006*; *Soltanian, 2016*).

## IMMUNOLOGY

Studies of ranaviral host immunity and immune evasion in amphibians are extensive, while similar work in reptiles is limited (*Grayfer et al., 2015*). The immunology section in the 2015 *Ranavirus* book, although comprehensive on amphibians, only mentions reptiles in passing (*Grayfer et al., 2015*). Immunology is an area in which a great number of unknowns remain for ranaviruses and reptiles.

### Innate

Antimicrobial peptides (AMPs) are likely involved in amphibian ranaviral defence. Amphibian antimicrobials such as E2P and R2P are cable of inactivating ranaviral virions through direct interaction at all temperatures tested (0–26 °C) (*Chinchar et al., 2001*). Reptile species also possess a range of antimicrobial peptides, primarily cathelicidins and $\beta$-defensins (*Preecharram et al., 2010*; *Van Hoek, 2014*; *Ageitos et al., 2017*). Homologs of the anti-ranaviral peptides in amphibians (class-four AMPs) have not been found in reptiles, although defensin-like peptides from the albumin of marine turtles possessed

antiviral activity against enveloped rhabdoviruses (*Chattopadhyay et al., 2006*). No reptilian AMPs have, however, been specifically assayed for anti-ranaviral activity.

Few studies have looked at the role of cytokines against ranaviruses in reptile immunity and these should be investigated in future studies. One study found that IFN-γ appears to have some antiviral activity in ranavirus infected soft-shelled turtle cells, although the mechanisms are unclear (*Fu et al., 2014*).

The reptile serum complement system also deserves further consideration, as it is capable of inhibiting viral replication (*Merchant et al., 2005*). Serum from American alligators (*Alligator mississippiensis*) exhibits antiviral activity against human immunodeficiency virus type-1, which has been attributed to action of the complement system (*Merchant et al., 2005*). The effect of the reptilian complement system on ranaviral replication efficiency has not been investigated.

Extensive work has attempted to elucidate the complex role of amphibian macrophages in ranaviral infection, although work in reptile hosts is limited (*Grayfer et al., 2015*). It has been hypothesized that ranaviral infection is partly dependent on the phagocytic and endocytic activity of macrophages. Ranaviruses overcome the antiviral defences of macrophages and use the cells for persistence and dissemination throughout the host. Ectothermic vertebrates, including reptiles, possess a unique type of phagocytic B cell capable of ingesting foreign particles (*Zimmerman et al., 2010*). It is conceivable that these phagocytic B cells may also be involved in ranavirus dissemination.

## Adaptive

Much less is known about the reptilian adaptive response than the innate response system (*Rios & Zimmerman, 2015*). Studies of the role of the adaptive immune system in clearing ranaviral infection have been almost exclusively restricted to amphibians and fish (*Chen & Robert, 2011*; *Grayfer et al., 2015*). The only studies of ranaviruses and the adaptive arm of the reptilian immune system have been through epidemiological studies. Anti-ranaviral IgY is produced as a long-lasting and specific adaptive response to infection and is the preferred target of reptilian serological assays (*Johnson et al., 2010*; *Zimmerman, Vogel & Bowden, 2010*; *Ariel et al., 2017*). The virus neutralising ability of anti-ranaviral antibodies detected in reptile populations has not been determined. Studies of T cell proliferation in response to ranaviral infection have not been conducted in reptiles and it is not clear if reptiles develop long-lasting immunological memory against ranaviral infection. Amphibian researchers have made a start on these questions, providing useful guidance for future studies in reptiles (*Grayfer et al., 2015*).

## TREATMENT

For treatment of acute ranaviral infection, several antivirals have been considered and tested (*Allender, 2012*; *Li et al., 2015*; *Sim et al., 2016*). However, there are few examples of their successful use to treat clinical cases (*Johnson et al., 2010*; *Allender, 2012*; *Miller et al., 2015*). Many *in vitro* antiviral studies that show promising results do not carry to *in vivo* models or have not been thoroughly tested *in vivo*.

Acyclovir, the most extensively studied antiviral in reptiles, does not appear to be an effective anti-ranaviral agent. Viral thymidine kinase (present in some herpesviruses and ranaviruses) is required for activation of acyclovir, which then blocks viral DNA replication through competitive inhibition of the viral DNA polymerase (*Beutner, 1995*). *In vitro* results have been mixed; *Johnson (2006)* found that acyclovir provided a dose-dependent partial inhibition of a FV3-like ranavirus, and *Ferguson et al. (2014)* found no statistically significant effect of acyclovir on FV3 replication. Plasma concentrations of orally dosed acyclovir do not reach levels in turtles that have been suggested as sufficient for ranavirus inhibition (*Allender, 2012*; *Gaio et al., 2007*). It is difficult to interpret the results of the use of this drug in uncontrolled clinical settings; however, it is clear that in several cases acyclovir has not stopped the progression of reptilian ranaviral disease (*DeVoe et al., 2004*; *Johnson et al., 2008*).

Pharmacological studies of the effectiveness of different antivirals at different severities and durations of ranaviral infection in reptiles have not been conducted, but would be extremely useful for guiding the treatment of acute ranaviral infection in reptiles.

Iridoviral vaccine development has been limited to the aquaculture industry (*Miller et al., 2015*). Frogs can produce long-lasting FV3-specific neutralising antibodies on second exposure (*Maniero et al., 2006*), suggesting it would be possible to develop vaccines for them. Reptiles can produce anti-ranaviral antibodies during infection (*Ariel et al., 2017*; *Johnson et al., 2010*), and vaccines have been developed for other reptilian pathogens with varying success (*Horner, 1988*; *Jacobson et al., 1991*; *Mohan et al., 1997*; *Marschang, Milde & Bellavista, 2001*; *Yang, Pan & Sun, 2007*). Vaccine research and development are extremely costly, and more epidemiological research is required to determine if the development of a ranaviral vaccine would be efficacious for wild reptilian populations. However, there are several instances where a vaccine could be useful for small scale use. For example: in zoo collections, for valuable broodstock, and for endangered or at risk populations. Epidemiological studies may feasibly identify and prophylactically treat animals most at risk.

Environmental temperature can have a substantial effect on the humoral immune system (e.g., antibody production) of ectotherms (*Tait, 1969*), which opens up the possibility of influencing the outcome of an infection *via* control of environmental temperatures (see Susceptibility section). Increased ambient temperature has been suggested as a treatment method for ranavirus infection in reptiles (*Hyndman & Marschang, 2017*). However, *Allender et al. (2018)* recently reported that increased temperature (22 °C to 27 °C) resulted in reduced median survival time of ranaviral infected Testudines. It is likely that there is a threshold temperature, which dramatically improves survival as occurs with chytridiomycosis in amphibians (*Berger et al., 2009*). Further investigation is required to determine the optimal temperature for increasing survival of ranaviral infected reptiles, which may also be viral and host species dependent.

## FUTURE RESEARCH AND CONCLUSIONS

The field of ranavirus research is dominated by studies on fish and amphibians, these studies can serve as a guide for the tremendous number of directions ranaviral research in reptiles

could take. An increase in the number of epidemiological studies and surveys of ranaviruses in reptile populations is required to understand the distribution of these viruses in the class Reptilia, and to identify at-risk populations. Pathogenesis and transmission of ranaviruses in reptiles are still poorly understood and will require elucidation before this disease can be correctly modelled and appropriately managed in reptile populations. Reptile ranaviral host immunity and immune evasion strategies of the virus are also under-represented in the literature. From predator to pollinator to prey, reptiles play vital roles in the ecosystems they inhabit, but like amphibians, reptiles are experiencing global declines (*Gibbons et al., 2000*). It is, therefore, imperative that research continues to expand our understanding of reptiles and ranaviruses to help protect this valuable part of biodiversity.

### Funding
The authors received no funding for this work.

### Competing Interests
The authors declare there are no competing interests.

### Author Contributions
- Wytamma Wirth conceived and designed the experiments, performed the experiments, analyzed the data, contributed reagents/materials/analysis tools, prepared figures and/or tables, authored or reviewed drafts of the paper, approved the final draft.
- Lin Schwarzkopf, Lee F. Skerratt and Ellen Ariel conceived and designed the experiments, authored or reviewed drafts of the paper.

### Data Availability
  The research in this article did not generate any data or code. This is a review article.

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
