# Peer review of "Ranaviruses and reptiles"

_PeerJ, doi:10.7717/peerj.6083_

## Round 0.1 · original submission · Major Revisions

There have been some substantial and valid recommendations made by the reviewers, adding some more detail in places and more comprehensive citations.

·

Basic reporting

Lines 391: This is an awkward fist sentence. I suggest eliminating it and beginning with the second sentence currently on line 392.

Line 45-46: I suggest changing “noticed” to “reported” and adding Duffus et al. (2015), “Distribution and Host Range of Ranaviruses” in Ranaviruses, as a citation.

Lines 176-177: This sentence may be better placed in introduction.

Lines 37-42: This section of the introduction seems simplistic. I think the brief description of non-avian reptiles is unnecessary. As the author’s note, most ranavirus research has been conducted on amphibians and the purpose of this manuscript is to summarize reptilian ranavirus research. In this light, I think the discussion on lines 85-90 would work better as part of the introduction.

Lines 351 – 353: See also: Echaubard et al., 2010; St- Amour et al., 2008; Haislip et al., 2012; Reeve et al., 2013, Forson and Storfer 2006; Kerby and Storfer 2009.

Experimental design

No Comment

Validity of the findings

Lines 321-324: Although some research had shown ranaviruses can persist and remain infective in, there are other investigations that indicate that ranaviruses degrade and lose infectivity rapidly in ecologically relevant conditions, perhaps limiting effective transmission to direct contact. Brunner et al. (2015) summarize this in “Ranavirus Ecology and Evolution: from Epidemiology to Extinction: in Ranaviruses.

Additional comments

Lines 203-230: Is it possible to add general patterns of susceptibility to this section or perhaps add a susceptibility own section? Brunner et al. (2015) state, “Until the recent studies by Brenes et al. (2014a ,b) comparative studies involving reptiles were generally lacking.” Has this changed in the three years since the publication of Ranaviruses? If certain clades are more susceptible, even at broad scales, it would be a valuable addition. Alternately, does ecology influence susceptibility; i.e., are aquatic and semiaquatic reptiles less susceptible than terrestrial reptiles due to evolutionary history with ranaviruses? Even if this information is sparse or unknown, it would be a valuable inclusion as a future research direction.

Lines 386-388: I recommend a sentence or two explaining a little more about this relationship, particularly the hypothesis that ranavirus may use macrophages as a route of infection. See Grayfer et al. (2015) “Ranavirus Host Immunity and Immune Evasion” in Ranaviruses.

Line 140-142: I recommend briefly describing the differences, pros, and cons, particularly between swabs and blood samples. It would be valuable for researchers designing a sampling method to have these comparisons in once place.

·

Basic reporting

In a few cases, references are missing for specific statements, and a few additional references are suggested.

Experimental design

This is not an original primary research paper. It is a review of relevant literature with a specific focus.

Validity of the findings

As a review of literature, some of the standards do not apply. Based on the literature search, some of the conclusions are somewhat speculative.

Additional comments

Ranaviruses are important pathogens in a wide variety of ectothermic vertebrates. As the authors correctly note, work on these viruses in reptiles has lagged behind studies in amphibians. The authors provide an overview of recent literature on this topic as well as pointing out areas in which specific studies are still very much needed. While it provides an overview of the subject, the authors provide limited detail, and in some cases overinterpret the available data (or the lack thereof, e.g. when discussing the uses of antivirals and vaccines). The paper does not present novel information or research, but could be of interest to people embarking on studies in this field as well as a basis for discussion on future research projects. There are a number of specific points that need to be addressed before the manuscript can be considered for publication. The research aspect of the paper relies on a literature survey of the past decade comparing references to reptiles and amphibians in the ranavirus literature. Is there a reason that fish were left out of this survey? It would be interesting to see if the trend to more mentions of reptiles in titles and abstracts corresponds to a trend to more mentions of the broad host range of these viruses in general.
More specific comments on the text:
Taxonomy
Lines 68-70: This section does not reflect the full current taxonomy of the ranaviruses, which also still include Santee-Cooper ranavirus and Singapore grouper iridovirus, which to not group in the categories listed.
Line 71: “All of the viruses that infect reptiles”: As the authors have noted, this is an expanding field, it would be better to qualify this, e.g. found so far and described in the literature… The authors further state that ranaviruses found in reptiles all belong to the CMTV-like and FV3-like groups, however, cases of ATV-like virus infections in reptiles are mentioned in Duffus et al. 2015 (in those cases ECV was described).
Diagnostics and Surveillance
Lines 121-123: This section on antigen detection is unfortunately placed between two sections on antibody detection. Please rearrange the text to be more consistent.
Lines 125-133: This section remains very theoretical, while none of the actual surveys looking for antibodies against ranaviruses in reptiles are cited, e.g. Johnson et al., 2010 and Ariel et al., 2017. While these authors are cited further above, but the text never actually mentions what was found.
Lines 142-144: Be more specific. What samples? What tissues?
Lines 135-150: The authors should also discuss the various specific PCRs that have been used for ranavirus detection, as these can have different targets and different specificities (e.g. only FV3-like viruses vs. all ranaviruses).
Line 153: The authors refer to “these methods” but have not yet specified any methods.
Lines 159-163: Again, the authors could be more specific as to what has been detected and what information has been gained as suggested using these methods in reptiles.
Lines 165-174: The suggested methods have not yet been sufficiently critically discussed to evaluate how useful they would be in specific situations as suggested. What methods in particular are the authors suggesting, or are they suggesting the development of new methods, and if so, which ones and why?
Distribution and host range:
Line 176: While the distribution of ranaviruses around the globe range does theoretically overlap the geographic distribution of reptiles, this is not a helpful observation. For example, the knowledge of ranavirus distribution in Africa is almost non-existent, while the distribution of reptiles there is quite significant.
Line 180: The authors state that some reptile populations are naïve to these viruses. What populations would they not consider naïve and what evidence do they have to differentiate between naïve and non-naïve populations?
Lines 181-184: It is important to note that sampling protocol may strongly affect the results of such studies, and should be evaluated when drawing conclusions or making comparisons.
Distribution and host range:
Line 209: Do not use the word “population” when referring to a captive group. Also, the authors cite Alves de Matos et al. (2011) here, but that publication did study wild squamates. The text should be rewritten accordingly.
Line 227: The authors mention endothermy here, but it might be helpful to mention the role of temperature in virus growth in the introduction or taxonomy sections when talking about what classes of animals can be infected. This topic is also mentioned again at the end of the manuscript, but needs some context earlier for better understanding.
Pathology:
Lines 251-252: Please add references for these statements.
In general: Why are various transmission studies with reptiles (with one exception) not mentioned here?
Vectors
Possibly discuss this in conjunction with findings of virus in blood of infected animals as well as with transmission studies and the modes of transmission used.
Reservoirs:
Lines 321-330: See also Munro et al., 2016. Survival of frog virus 3 in freshwater and sediment from an English lake. J Wildl Dis 52:138-142. And: Johnson and Brunner, 2014. Persistence of an amphibian ranavirus in aquatic communities. Dis Aquat Org 111:129-138.
Treatment
In general, this section should be more cautious based on the lack of specific data. E.g. in the discussion of the use of antivirals, acyclovir was not shown to be effective against a ranavirus in vitro (Furguson et al., 2014 which is only available as a proceedings abstract under: http://members.arav.org/resource/resmgr/Files/Proceedings_2014/27.pdf, while Allender, 2012 discussed the possible use and pharmacokinetics, but did not demonstrate efficacy against ranaviruses.
Line 413 stating that „antiviral therapy is ineffective at reviving them“, possibly because animals are already in critical condition implies that an effective antiviral treatment is available, which is not currently the case. The same is true of line 417 „application of treatment….“
Lines 423-424: It might be more helpful to cite other examples of attempts to vaccinate reptiles against viruses, e.g. Jacobson et al. 1991 (rattlesnakes against paramyxoviruses, not successful), or cases in which crocodilians have been vaccinated against poxviruses and West Nile virus (see e.g. Horner, 1988). In this section it might also be helpful to mention possible carrier states in conjunction with vacicnation as a cautionary note.
Line 447: It would be helpful to include references for global declines of reptiles if this is mentioned as a driving force for continued research on this topic.
Figure 1: for interpretation of this graph, it would be helpful to note the number of publications taken into consideration again in the caption.
Table 1: I am confused as to how specific publications were chosen for this as it is not (by far) comprehensive. This needs either some explanation or needs to be significantly expanded.
Additional minor points:
When using taxonomical terms including family names in virology, these should be italicized (e.g. Iridoviridae).
The term Ranavirus (italicized, capitalized) refers to a taxonomic construct (genus), do not capitalize or italicize the word when referring to a specific virus.
When discussing the taxonomy of the hosts in which ranaviruses have been detected (in the abstract and the introduction), remember that lizards are not monophyletic.
Common names should not be italicized (see Burmese star tortoise, line 188)
The use of the term “turtle” vs. “tortoise” should be considered – in general, terrestrial turtles/tortoises are referred to as tortoises in American and British usage (see line 189 – Hermann’s turtles are usually referred to as tortoises in the USA and the UK).

Reviewer 3 ·

Basic reporting

This review article intends to summarize the research progress in relation with ranaviruses and reptiles. It seems likely that the article covers a wide range of aspects; but it really contains rather less information. For example, when you read the abstract, only the last sentence gives a bit information about what is presented in this article. In immunology related section of this article, only a few AMPs and IFN-gamma are mentioned, with the inclusion of only two Ig isotypes. In fact, there are quite a few other reports on immune system of reptiles, including turtle and lizard, and it is not corret to say that there only two types of Igs in reptiles.

Experimental design

It might be important to summarize various aspects in relation with ranavirues and reptiles.

Validity of the findings

Research progress is not well or deeply reflected in the article.

Additional comments

A detailed digging of data may be required to reflect the research progress in relation with reptiles and ranaviruse. There are more published articles to be cited.

---

## Round 0.2 · Minor Revisions

A number of clarifications have been suggested by Reviewer 2, please address these and correct typos that have been indicated.

·

Basic reporting

No comment

Experimental design

No comment

Validity of the findings

No comment

Additional comments

The author's have addressed my initial comments and generally improved the overall quality of the manuscript. I believe it could be a valuable overview for researchers beginning investigations on ranavirus infections in reptiles.

·

Basic reporting

The manuscript is much improved. It currently contains multiple typos and careless mistakes and should be carefully edited before final publication.

Experimental design

As previously stated

Validity of the findings

Much improved.

Additional comments

Specific comments:
Abstract:
Why did you remove “known to be” from the sentence “More recently, reports of ranaviral infections of reptiles are increasing with over 12 families of reptiles currently known to be susceptible to ranaviruses, most of which are turtles and lizards.ranaviral infection”?

Diagnostics and surveillance
Page 5, paragraph IHC, last line: “indicating that viraemia proceeded the systemic infection observed in these animals.” Should be preceeded.

Distribution, host range and impact
- Testudines (turtles, tortoises and terrapins)
Johnson et al. 2007 also infected red eared sliders.
Trionyx sinensis is currently known as Pelodiscus sinensis

Distribution and host range - Squamata (lizards and snakes)
Final line . ranaviruses in wild squamates: add a reference to Alves de Matos et al., 2011 here.

Pathology:
Line: “challenge experiments, and the fact the reports are often confounded is co-infections”: This line needs some work I am not sure what the authors are trying to say here

Reservoirs
and soil (even others replies.....: should be even other reptiles

Immunology - Innate
work in reptile hosts if limited: is limited
Ranavirus overcomes the antiviral defenses of macrophages and uses the cells: should be Ranaviruses overcome….and use…

Immunology - adaptive
Ranaivurs IgY: Typo, and these are not ranavirus antibodies, but antibodies against ranaviruses

Treatment
De Veo: should be written DeVoe (also in references)
Further investigation is required to determine the optimal temperature for increasing survival from aof ranaviral infectioninfected reptiles.: It would be good to mention here that this will likely depend on the host speices.

Table:
What does H stand for?

Reviewer 3 ·

Basic reporting

My comments have been met, and I do not have further comments.

Experimental design

N/A

Validity of the findings

N/A

Additional comments

no further comments.

---

## Round 0.3 · accepted · Accept

Please correct: "PCR based assays have been used..." to "PCR-based assays have been used...".

#